# Single-shot ultrafast terahertz photography

Junliang Dong [1,2] ✉, Pei You[1,2], Alessandro Tomasino [1,2], Aycan Yurtsever[1] & Roberto Morandotti [1] ✉

Multidimensional imaging of transient events has proven pivotal in unveiling many fundamental mechanisms in physics, chemistry, and biology. In particular, real-time imaging modalities with ultrahigh temporal resolutions are required for capturing ultrashort events on picosecond timescales. Despite recent approaches witnessing a dramatic boost in high-speed photography, current single-shot ultrafast imaging schemes operate only at conventional optical wavelengths, being suitable solely within an optically-transparent framework. Here, leveraging on the unique penetration capability of terahertz radiation, we demonstrate a single-shot ultrafast terahertz photography system that can capture multiple frames of a complex ultrafast scene in non-transparent media with sub-picosecond temporal resolution. By multiplexing an optical probe beam in both the time and spatial-frequency domains, we encode the terahertz-captured three-dimensional dynamics into distinct spatial-frequency regions of a superimposed optical image, which is then computationally decoded and reconstructed. Our approach opens up the investigation of non-repeatable or destructive events that occur in optically-opaque scenarios.

Single-shot ultrafast photography[1–3] has emerged as a key technique to elucidate the complex dynamics underlying miscellaneous ultrafast phenomena in nature. Propelled by recent advances in the fields of ultrafast lasers, high-speed cameras, and computational imaging, single-shot ultrafast optical imaging has been able to capture two-dimensional (2D) transient scenes at 100 billion frames per second[1], fast enough to visualize optical pulses traveling through space at the speed of light. This has the potential to revolutionize a number of fascinating applications where, e.g., ultrafast light pulses can either hide visible objects by means of optical cloaking[4] or bypass and unveil them using self-accelerating beams[5]. In addition, single-shot ultrafast optical imaging allows the recording of non-repeatable or destructive events, such as explosions[6], laser chaos[7], and irreversible chemical reactions[8], outperforming conventional pump-probe methods. However, to date, state-of-the-art single-shot ultrafast imaging techniques have only been experimentally demonstrated within the optical window[9]. This restriction prevents such techniques from exploring many critical ultrafast phenomena occurring in media with a short optical penetration depth, such as the dynamics of laser ablation in ceramics[10], magnetization in iron films[11], and carrier excitations in semiconductors[12].

Recently, imaging using terahertz (THz) radiation[13–17] (electromagnetic waves with wavelengths typically ranging from 30 μm to 3 mm) has garnered significant interest due to its ability to 'see through' various materials. THz waves experience considerably weaker absorption in metals, semiconductors, and dielectrics compared to ultraviolet, visible, and infrared light[18]. This high penetration power of THz waves represents a distinct advantage, making it suitable for probing relatively thick, multilayered structures[19], where optical probes cannot be used. In addition, unlike X-rays, THz radiation has a low photon energy (~meV), which does not cause deleterious effects in biological tissues[20]. Moreover, THz imaging also allows spectroscopic identification of various substances, as key fingerprints of many chemical compounds are located in this frequency range[18]. Therefore, ultrafast imaging with THz radiation can be an effective way to capture ultrafast scenes that are not accessible via optical frequencies, even in multiple degrees of freedom (hyperspectral imaging). Despite such attractive properties, single-shot ultrafast THz imaging is still in the embryonic stage, due to the absence of high-speed THz cameras[21].

[1]Institut national de la recherche scientifique, Centre Énergie Matériaux Télécommunications, Varennes, QC J3X 1P7, Canada. [2]These authors contributed equally: Junliang Dong, Pei You, Alessandro Tomasino. ✉e-mail: Junliang.Dong@inrs.ca; Roberto.Morandotti@inrs.ca

Some holography-based approaches have been theoretically proposed[22,23], however, these methods may be difficult to implement experimentally and are only limited to simple scenes, such as a focused THz beam spot.

Here, we propose and demonstrate a single-shot ultrafast THz photography system by exploiting the electro-optic sampling (EOS) technique for THz detection using an optical probe beam multiplexed in both the time and spatial-frequency domains. According to the EOS mechanism[24,25], THz waveforms can be coherently reconstructed by probing the birefringence induced by the THz electric field into a Pockels crystal. Since the probe beam typically features a wavelength in the near-infrared region, it is possible to use a conventional charge-coupled device (CCD) camera to map THz-induced variations in probe polarization, revealing a 2D THz image[26]. We find that multiplexing the probe beam enables us to encode the evolution of a complex ultrashort event captured by a THz wave into different and uniquely marked spatial-frequency regions in Fourier space. This leads to the formation of a multiplexed image acquired by a CCD camera, which can be processed to recover the consecutive frames corresponding to the 2D transient scenes, featuring sub-picosecond ($<10^{-12}$ s) interframe time intervals.

## Results

### System design

The basic principle of probe-beam multiplexing[27,28] is illustrated graphically in Fig. 1. When an object is uniformly illuminated, the band-limited power spectrum of the acquired image only occupies the central region around the Fourier domain origin[27], as shown in

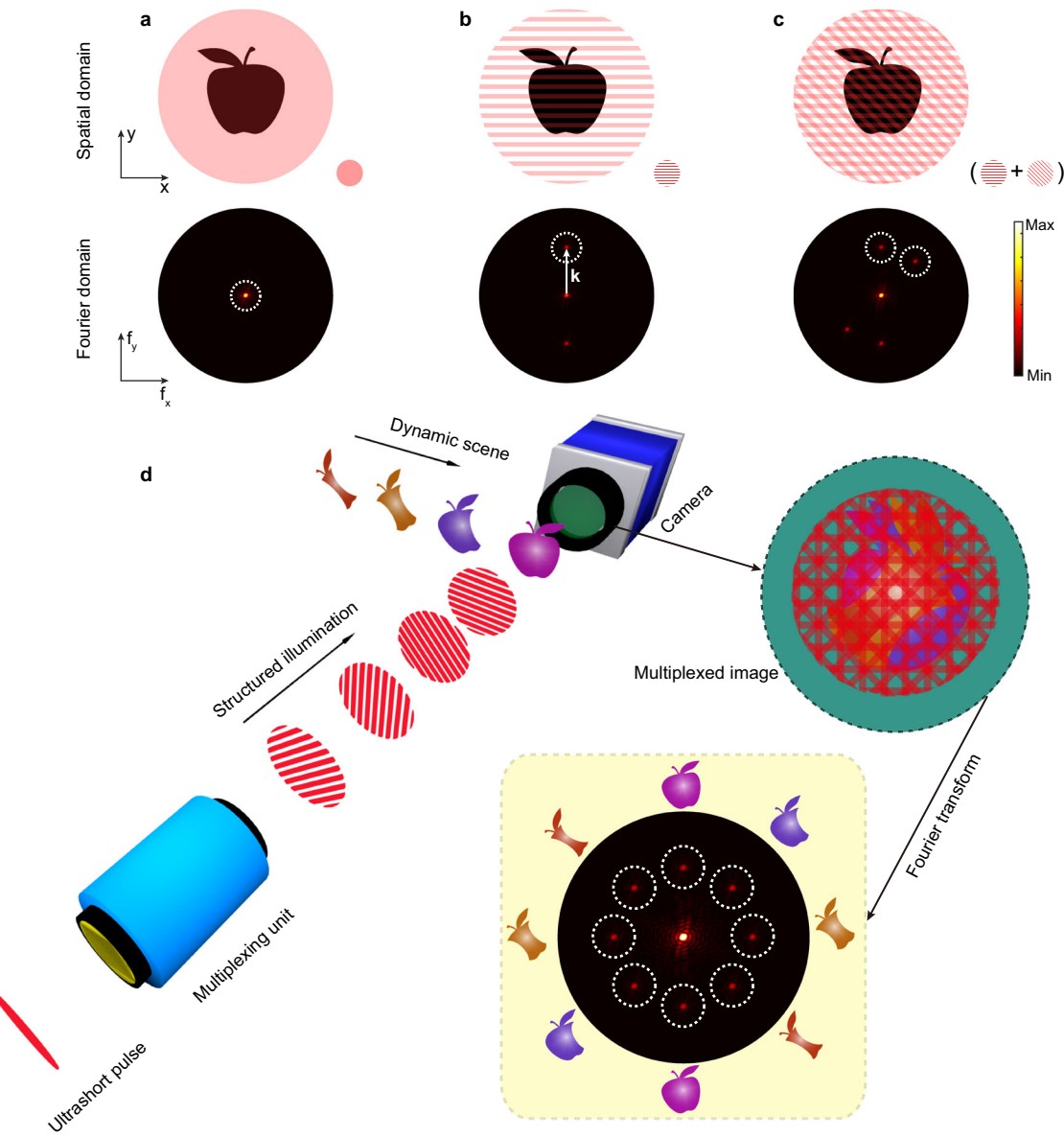

**Fig. 1 | Concept of probe-beam multiplexing in the spatial-frequency and time domains. a** An object (here, an apple) is uniformly illuminated in real space. The frequency content of the resulting image is band-limited and is mainly localized in the central region of Fourier space. **b** Illuminating the same object with a sinusoidal intensity modulation produces two 'image copies' of the object in two unexploited regions, symmetrically located in Fourier space with respect to the origin. Their actual locations (i.e. the distance from the origin) depend on the spatial-frequency **k** of the modulation pattern. **c** By varying the pattern orientations, 'image copies' can be moved to different non-overlapping regions in the Fourier domain, thus allowing for the implementation of a multiple-illumination scheme without any mutual interference. **d** By judiciously implementing multiple spatially modulated and temporally delayed patterns, a dynamic scene can be imaged. Despite being overlapped in the multiplexed image captured by the camera, these frames are well separated in Fourier space, and can be extracted and recovered computationally.

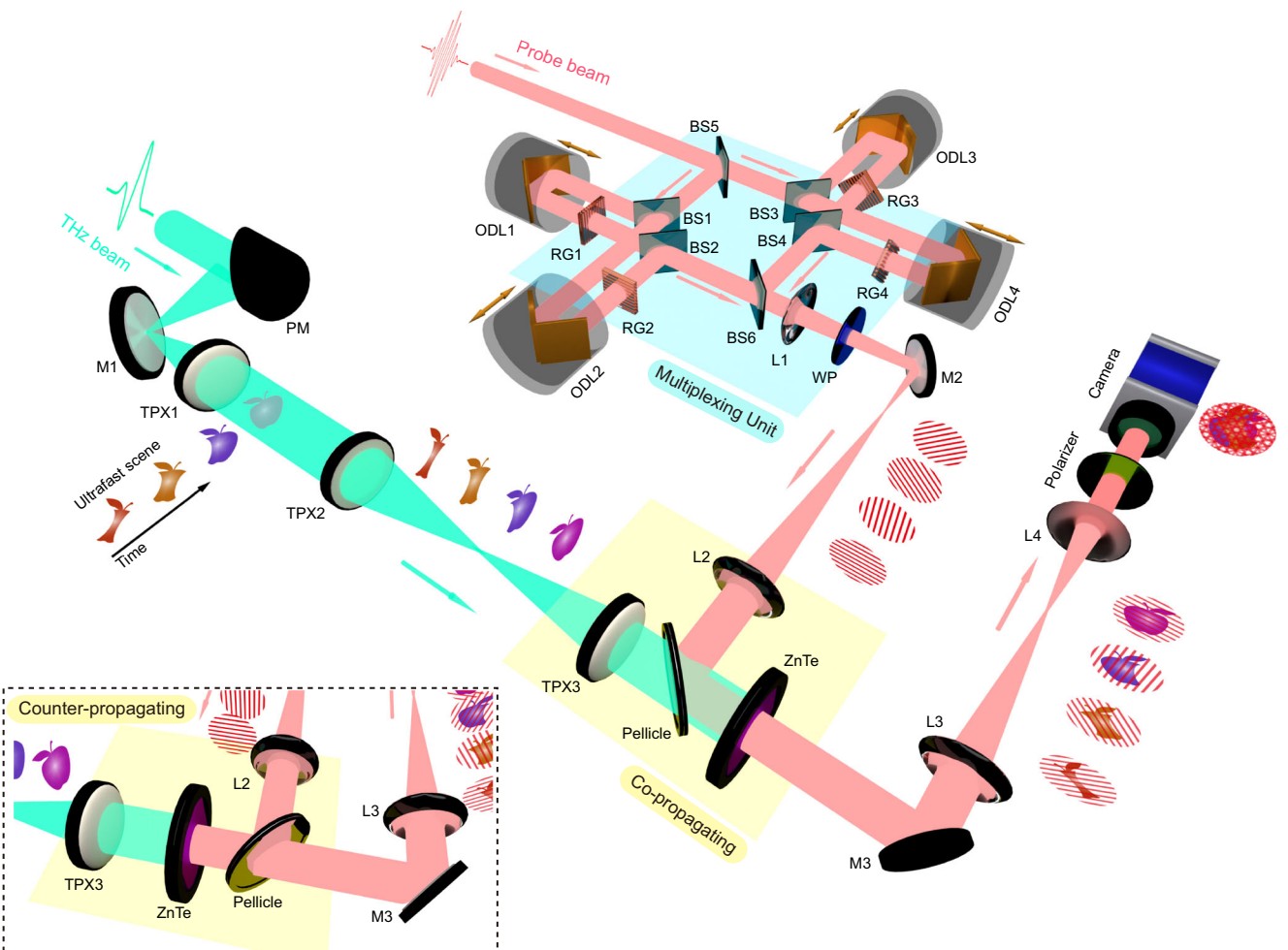

**Fig. 2 | Schematics of the proposed single-shot ultrafast THz photography system.** Details about the experimental setup are introduced in the Method section. The EOS technique can be operated in either a co-propagating or counter-propagating configuration, depending on the specific experiment requirements. In the counter-propagating configuration (see inset), the multiplexed probe beam is sent to the rear of the ZnTe crystal, traveling in the opposite direction to the THz beam. After being reflected from both surfaces of the crystal, the multiplexed probe beam then co-propagates with the THz pulse. The beam portion reflected from the left surface of the crystal is modulated by the THz electric field, and thus, carries the 2D information of the ultrafast scene. BS1-BS6 beam-splitter, ODL1-ODL4 optical delay lines, RG1-RG4 Ronchi gratings, L1-L4, lens; M1-M2 mirrors, PM parabolic mirror, TPX1-TPX3 THz lens, WP quarter-wave plate.

Fig. 1a. If the same object is illuminated using a sinusoidal modulation with a spatial-frequency vector **k**, a pair of 'image copies' of the original object will be created at $\pm\mathbf{k}$ in the Fourier domain, see Fig. 1b. As demonstrated in Methods, it is possible to further extend this concept and multiplex in the spatial-frequency domain by exploiting modulation patterns with several distinct orientations, see Fig. 1c. Since there is still no overlap among the image copies in the Fourier domain, any one of them can be used to recover the original image of the object. This strategy allows imaging of a complex transient scene by means of multiple spatially modulated and temporally delayed illumination patterns, as illustrated in Fig. 1d, which we can imprint into the probe beam via the EOS technique.

Figure 2 shows the schematics of the proposed single-shot ultrafast THz photography system. The system is driven by an 800 nm femtosecond laser, which delivers both pump and probe beams with a pulse duration of 150 fs. A collimated THz beam, generated from a LiNbO$_3$ crystal, passes through an ultrafast scene and is then detected via the EOS technique implemented in a ZnTe crystal[29]. The initial probe beam is split into four equally intense sub-pulses using a multiple beam-splitting configuration. Each sub-pulse is guided into an optical delay line to control its arrival time (for multiplexing in the time domain). After being delayed, the sub-

pulses are sent through four Ronchi gratings, each of which is oriented along a different angle (for multiplexing in the spatial-frequency domain). These four sub-pulses are subsequently recombined through the last beam splitter to form the final probe beam. The ±1 diffraction orders of the beam are collected, forming sinusoidal modulation patterns in the probe profile. The resulting multiplexed probe beam, consisting of four sub-pulses with different modulation patterns and time delays, illuminates the detection crystal by co-propagating (or counter-propagating) with the THz wave that carries the temporal evolution of the ultrafast scene. The polarization of each sub-pulse is modulated by the THz electric field via the EOS technique, producing a frame of the ultrafast scene at a specific time. All frames obtained in a single-shot manner are stacked into a single multiplexed image, which is finally captured by the CCD camera. Despite being overlapped in the spatial domain, these frames are well separated in Fourier space and can be thus extracted and recovered. The inter-frame time intervals are flexible and can be easily adjusted by varying the optical delay lines. For simplicity, we limit the total number of frames to four, but this can be increased by inserting additional pairs of beam-splitters and optical delay lines (see discussions about the maximum number of frames that can be achieved in Supplementary Information).

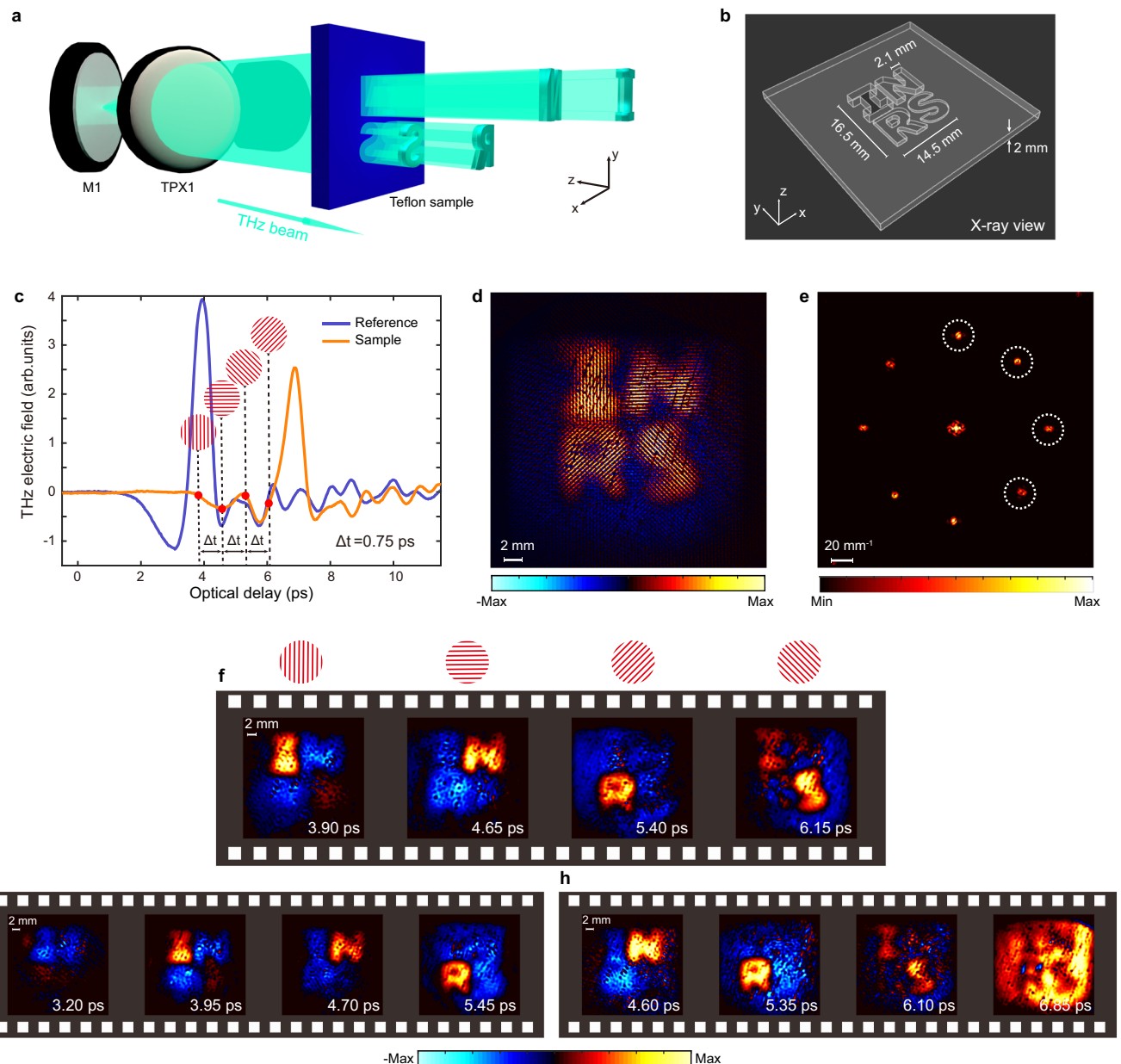

**Fig. 3 | Ultrafast imaging of a THz pulse propagating through a patterned structure. a** The dynamic scene consists of a collimated THz beam traveling through a Teflon sheet (thickness: 2 mm) with four engraved letters ('I', 'N', 'R', and 'S'). **b** Detailed geometries of the letters shown in an 'X-ray' view. The stroke width of the letters is ~2.1 mm. The engraving depths of the four letters, 'I', 'N', 'R', and 'S', are 2 mm, 1.5 mm, 1.0 mm, and 0.5 mm, respectively. As a result, the letter 'I' is engraved throughout the sample along the depth direction. **c** THz waveforms acquired via the EOS technique with and without the sample. The arrival times of the four sub-pulses for imaging are highlighted with red dots. The inter-frame time intervals are all equal and set to 0.75 ps. **d** Multiplexed image acquired by the camera in real space. **e** 2D Fourier transform of the image in **d**. The 'image copies' of the four frames to be extracted are indicated by circles. **f** Recovered frames from the multiplexed image in **d**. **g** Recovered frames captured when the probe beam arrived 0.70 ps earlier relative to the case in **f**. **h** Recovered frames captured when the probe beam arrived 0.70 ps later relative to the case in **f**. The image contrast of the frames is given by the amplitude of the THz electric field.

## Experimental demonstration

To demonstrate the ultrafast imaging capability of our system, we first imaged a 'light-in-flight' scene by capturing a THz pulse propagating through a Teflon sheet (which is opaque to optical frequencies) with engraved letters ('I', 'N', 'R', and 'S'; see Fig. 3a). The four letters, whose geometries are depicted in Fig. 3b, are engraved at different depths. Before imaging, the THz waveforms were recorded with and without the sample as plotted in Fig. 3c. By referring to the temporal evolution of the THz waveforms, we set the arrival time of the multiplexed probe beam to 3.90 ps with an inter-frame time interval of 0.75 ps, in order to capture representative frames of the THz waves propagating through the object. Figure 3d shows the multiplexed image captured by the

camera, and Fig. 3e depicts its corresponding spectrum. It is apparent that the frequency content of each frame is well separated in the Fourier domain. Following post-processing (see Methods), the four frames were extracted and reconstructed individually. The recovered frames, corresponding to the 2D distribution of the THz electric field at four specific times, are shown in Fig. 3f. As expected, the THz pulse passed through the letter 'I' first, because THz waves travel faster in air than in Teflon. Following the 'I-N-R-S' sequence, the recovered frames display a dynamic scene when the THz pulse propagates through the letters one by one. Figure 3g, h display another two sets of representative frames captured when the probe beam was temporally shifted by ±0.70 ps. These time-lapse frames clearly reveal the

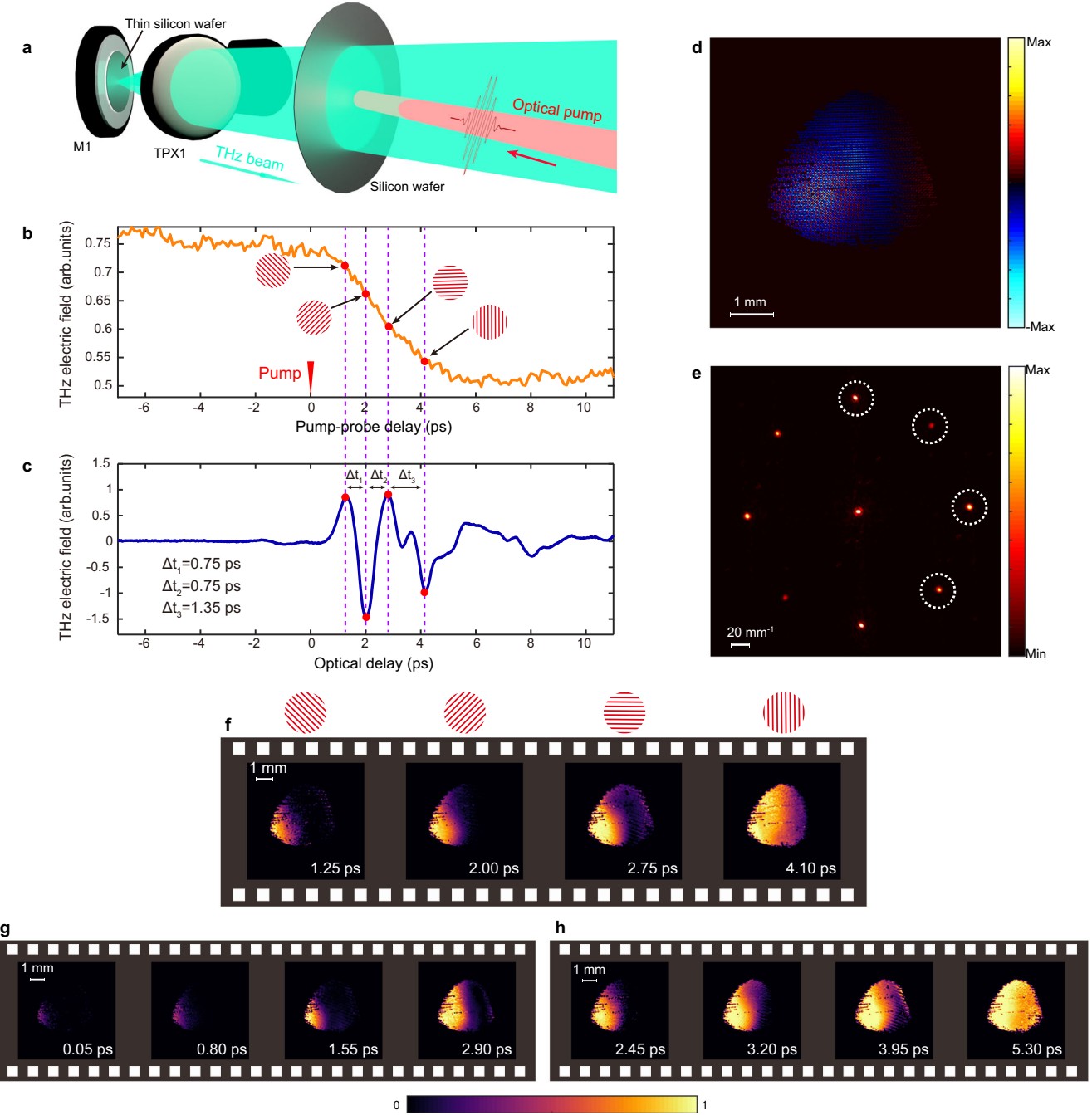

**Fig. 4 | Ultrafast imaging of the photo-excited carrier dynamics in silicon. a** The dynamic scene we imaged was the carrier generation in an undoped silicon wafer (thickness: 500 μm) pumped by a near-infrared laser pulse. The incident angle of the optical pump was -30°, with a diameter of ~2 mm and an average power of 50 mW. **b** Optical pump-THz probe measurements. Note that for negative pump-probe delay values, the optical pump arrives after the THz pulse. **c** Multi-cycle THz pulse used for ultrafast imaging. Such a pulse was obtained by attaching a thin silicon wafer (thickness: 35 μm) onto the mirror M1. The four sub-pulses in the multiplexed probe beam were temporally aligned with the four peaks of the multi-cycle THz pulse, in turn leading to inter-frame time intervals of 0.75 ps, 0.75 ps, and 1.35 ps, respectively. **d** Image acquired by the CCD camera when the arrival time of the multiplexed probe beam was set to 1.25 ps after photoexcitation. **e** 2D Fourier transform of **d**, with image copies from each sub-pulse circled. **f** Recovered frames from the multiplexed image in **d**. **g** Recovered frames when the optical pump arrived 1.20 ps later relative to the case in **f**. **h** Recovered frames when the optical pump arrived 1.20 ps earlier relative to the case in **f**. The imaging contrast plotted in the frames is the modulation ratio, calculated by normalizing the individual frames with a reference frame, which was taken when the carriers within the illumination region were fully excited (at a time instant larger than 6 ps after photoexcitation).

spatiotemporal evolution of a bipolar THz pulse traveling through the sample, as a result of distinct propagation speeds in different materials.

To explore the potential applications of our system in advanced materials, we visualized the ultrafast photoexcitation of carriers in bulk silicon. As shown in Fig. 4a, an additional near-infrared pump beam was made to obliquely impinge on the center of the silicon wafer. We first characterized the temporal dynamics by performing a conventional optical pump-THz probe measurement[12]. By fixing the time position of the THz pulse, the transmitted THz electric field was recorded as we scanned the arrival time of the optical pump. As shown in Fig. 4b, when the optical pump arrived at the same time as the THz pulse, we started

to observe a decrease in THz transmission, as photoexcitation led to carrier generation in the wafer, thus changing its conductivity[30]. The decrease lasted for ~6 ps, until the carriers were fully excited within the illumination area. In order to capture a 2D dynamic scene of a ~6-ps-long transient, we need a larger imaging time window corresponding to the THz pulse length. To this end, we attached a 35-μm-thick silicon wafer onto mirror M1 (see Fig. 4a). Multiple reflections within such a thin wafer prolonged the THz pulse duration, resulting in an imaging time window of several picoseconds, as shown in Fig. 4c. The four sub-pulses in the multiplexed probe beam were temporally aligned with the four peaks of the multi-cycle THz pulse, leading to inter-frame time intervals of 0.75 ps ($\Delta t_1$), 0.75 ps ($\Delta t_2$), and 1.35 ps ($\Delta t_3$). The first set of representative frames was captured when the arrival time of the multiplexed probe beam was set at 1.25 ps after photoexcitation. Figure 4d shows the multiplexed image obtained from the camera, while its Fourier transform is depicted in Fig. 4e. The frames recovered after post-processing are shown in Fig. 4f. The imaging contrast used here is the modulation ratio, as calculated by normalizing the acquired frames with a reference frame, which is the image taken when the carriers were fully excited (≥6 ps after photoexcitation). These four frames illustrate how the carriers were excited from left to right due to the oblique incidence of the optical pump. Figure 4g, h display two other sets of frames attained by shifting the arrival time of the optical pump by ±1.20 ps, while keeping the multiplexed probe beam fixed. The combination of all these frames provides a comprehensive view of the spatiotemporal dynamics associated with the excited carriers in silicon, upon illumination with a femtosecond laser pulse.

In conclusion, we have presented a single-shot ultrafast photography system that can capture ultrashort events in optically-opaque scenarios. Our single-shot method essentially bypasses the need for high-speed devices operating at THz wavelengths, yet it is powerful in providing the spatiotemporal evolution of a complex ultrafast scene with sub-picosecond resolution. The inter-frame time interval of our system is easily tunable and the temporal resolution is only limited by the probe pulse duration. When the frame interval is smaller than the pulse duration, the interference pattern resulting from two consecutive sub-pulses will impose an additional and uncontrollable modulation on the image. Such an undesired effect results in the appearance of spurious frequency content in the Fourier domain, in turn worsening the frame recovery procedure. As a final remark, the spatial resolution can be enhanced by exploiting a near-field configuration. A modified version of our system could potentially function as a single-shot ultrafast THz microscope by attaching the objects directly onto the detection crystal[31], thus allowing us to capture ultrafast scenes at the microscale. We envisage that our system will provide unprecedented insights into a broad variety of exotic dynamics that occur in advanced materials and structures, e.g. 2D materials[32,33] and even biological matter[34].

## Methods
### Imaging model
The electro-optic sampling (EOS) technique can accurately retrieve both the amplitude and phase of a THz electric field. The THz pulses induce birefringence in a Pockels crystal (such as ZnTe), which in turn causes a change in the polarization state of a co-propagating optical probe beam. By using a quarter-wave plate and polarizer, the change in polarization can be mapped onto a CCD camera as an intensity modulation. In general, the achieved image $P(x,y,t)$ is related to the THz electric field $E_{THz}(x,y,\omega_{THz},t)$ as follows[31]:

$$P(x,y,t) = \frac{1}{c}I_0(x,y,t)\omega L n_0^3(\omega)r_{14}E_{THz}(x,y,\omega_{THz},t), \quad (1)$$

where $c$ is the speed of light in a vacuum, $I_0(x,y,t)$ is the intensity distribution of the optical probe on the crystal upon uniform

illumination, $L$ is the thickness of the crystal, $\omega$ is the frequency of the probe, $n_0(\omega)$ is the refractive index, and $r_{14}$ is the electro-optic coefficient of the crystal.

In our configuration, the probe beam is multiplexed both temporally and spatially. Therefore, its intensity distribution $I(x,y,t)$ is modified as:

$$I(x,y,t) = \sum_{n=1}^{4} I_n\left(x,y,t - \triangle t_{Dn}\right)M_n(x,y), \quad (2)$$

$$I_n\left(x,y,t - \triangle t_{Dn}\right) = \begin{cases} 0, t < \triangle t_{Dn} \\ I_0(x,y,t), t \geq \triangle t_{Dn} \end{cases}, \quad (3)$$

where $\triangle t_{Dn}$ is the time delay induced by the $n$-th optical delay line. $M_n(x,y)$ is the spatial modulation pattern produced by the $n$-th Ronchi grating, which consists of periodic stripes. In principle, vertical stripes can be modeled as periodic square waves along the $x$-direction[28]:

$$M_n(x,y) = M_n(x+T,y) = \begin{cases} 1, 0 \leq |x| \leq T/4 \\ 0, T/4 < |x| \leq T/2 \end{cases}, \quad (4)$$

where $T$ is the period of the modulation pattern. By integrating Eqs. (2)–(4) into Eq. (1) and taking the spatial Fourier transform, the multiplexed image achieved via EOS in the spatial-frequency domain can be expressed as:

$$\widetilde{P}\left(k_x,k_y\right) = \sum_{n=1}^{4}\left[\widetilde{S}_n\bigotimes\sum_{m=-\infty}^{+\infty}\frac{2\sin(mk_0 T/4)}{m}\delta\left(k'_{xn} - mk_0\right)\right], \quad (5)$$

$$\widetilde{S}_n = \mathscr{F}\left[\frac{1}{c}I_n\left(x,y,t - \triangle t_{Dn}\right)\omega L n_0^3(\omega)r_{14}E_{THz}\left(x,y,\omega_{THz},t - \triangle t_{Dn}\right)\right]. \quad (6)$$

Here $\mathscr{F}$ represents the Fourier transform and $\bigotimes$ denotes the convolution operator. In Eq. (5), the right-hand side of $\bigotimes$ comes from the Fourier transform of the Ronchi grating. This is because multiplication of the images with the modulations in real space is equivalent to the convolution of their corresponding spectra in Fourier space. $k_0 = 2\pi/T$ is the fundamental spatial frequency. We define $k'_{xn} = k_x\sin\theta_n + k_y\cos\theta_n$, where $\theta_n$ is the rotation angle of the $n$-th modulation pattern. Based on Eqs. (5) and (6), the image obtained after EOS detection is the sum of four images. The Dirac delta function $\delta$ creates spectrum copies of the original image $\widetilde{S}_n$ and shifts them to a unique spatial-frequency location corresponding to the $n$-th modulation pattern.

### Imaging system
Our imaging system was driven by an 800-nm Ti:Sapphire amplified pulsed laser that delivered 2 mJ pulses with a 150-fs pulse duration at a 1-kHz repetition rate. The main optical pump and probe beams were obtained by means of a 90/10 beam-splitter. THz generation was based on the optical rectification technique, which was achieved in a 63°-cut LiNbO$_3$ crystal, relying on a tilted-pulse-front excitation scheme[35]. The wavefront of the pump beam was tilted using customized reflective grating with 1800 grooves/mm. The generated THz beam was first magnified by a factor of 10 via a pair of 90° off-axis parabolic mirrors and was then collimated to be employed as shown in Fig. 2. This provided a THz pulse train with a ~1.2-THz-wide spectrum (full-width-half-maximum) with a central frequency of ~0.5 THz. The THz beam was further guided by a series of components (PM, M1, and TPX1) and ultimately formed a collimated beam with a diameter of ~2 cm (measured at 10% of the THz field amplitude peak), impinging on the ultrafast scene to be recorded. The images carried by the THz wave were relayed to the detection crystal by two TPX lenses (TPX2 and

TPX3). The probe beam was sent to the multiplexing unit based on beam-splitter configurations, in which four identical Ronchi gratings (20 lp/mm) and optical delay lines (with a minimum step size of 25 fs) were used to multiplex the probe beam. The four Ronchi gratings were oriented at 0°, 45°, 90°, and 135°. Two imaging lenses (L1 and L2) were used to form sinusoidal fringe patterns on the detection crystal. To do this, the distance between lens L1 and each grating was equal to the focal length for each sub-pulse. The THz pulses were characterized using the EOS technique in a 1-mm-thick ZnTe <110> crystal with dimensions of 1 cm by 1 cm. For the 'light-in-flight' experiment, EOS was conducted in the counter-propagating configuration because the THz image of the object needed to be shrunk down in order to be imaged on the relatively small ZnTe crystal area (TPX2 focal length: 100 mm; TPX3 focal length: 35 mm; L1 focal length: 250 mm; L2 focal length: 500 mm). The Teflon sample was fabricated using the computer numerical control (CNC) machining technique. For the carrier dynamics experiment, EOS was used in a co-propagating configuration (TPX2 focal length: 100 mm; TPX3 focal length: 65 mm; L1 focal length: 200 mm; L2 focal length: 250 mm). The near-infrared pump beam used for carrier excitation in the silicon wafer was obtained from the main pump beam using a 90/10 beam-splitter, and its arrival time was controlled by another optical delay line (with a minimum step size of 25 fs). The multiplexed image was captured by a CCD camera (pco.pixelfly) featuring a total size of 1392 × 1040 pixels and a resolution depth of 14 bits. The pixel size was 6.45 μm × 6.45 μm. The polarizer in front of the CCD camera was oriented to detect both the positive and negative THz electric field transients, thus leading to the formation of bipolar frame images. The estimated spatial resolution of the imaging system was about 0.56 mm, as dictated by the central frequency of our THz source (see Supplementary Fig. 3).

## Post-processing procedure

Post-processing is used to recover the individual frames, as graphically illustrated in Supplementary Fig. 2. To start, a 2D Fourier transform was performed on the multiplexed image captured by the camera. One image copy in the Fourier domain (corresponding to the frame to be recovered) was isolated using a spatial-frequency band-pass filter, which removes all the other image copies. In our study, a 2D band-pass filter with a rectangular window was used to extract the image copy. The width of the window (in pixels) influences the spatial resolution of the recovered frame. The larger the window width, the more the frequency content is preserved, because any spatial frequency information outside the window is discarded. It is important to note that the width of the filter should also be small enough to prevent interference or crosstalk with image copies in the adjacent frames. The effect of filter width is investigated in detail in Supplementary Fig. 3. For our 'light-in-flight' experiments, the width of the filter was 80 pixels, and for the carrier dynamics experiment, the width was 140 pixels. Once the modulation frequency was determined, the isolated spatial-frequency content was digitally transferred to the Fourier domain origin, in turn transforming the sinusoidal modulation into uniform illumination. Finally, by applying the inverse Fourier transform on this filtered and rearranged spectrum, the original frame that was encoded in the shifted region in the Fourier domain could be recovered.

## Data availability

All experimental raw data generated in this study are provided in the Source Data file. All the other relevant data are available from the corresponding authors upon request. Source data are provided with this paper.

## Code availability

Any simulation and computational codes for this study are available from the corresponding authors upon request.

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

## Acknowledgements

This work was supported by the Natural Sciences and Engineering Research Council of Canada (NSERC) through the Discovery, Strategic, and Explorations grant programs, and from the Fonds de recherche du Québec - Nature et technologies (FQRNT) through the Audace Program. J.D. acknowledges financial support from a Mitacs Elevate Postdoctoral Fellowship. R.M. would also like to acknowledge support from the Canada Research Chair Program.

## Author contributions

J.D. conceived the idea and designed the experiments. J.D., P.Y., and A.T. established the imaging system, performed the measurements, and analyzed the experimental results. A.Y. contributed to the discussion of the results. R.M. supervised and coordinated the project. All authors contributed to the writing of the manuscript.

## Competing interests

The authors declare no competing interests.
