## [Peer Review File · Nature Communications]

Single-shot ultrafast terahertz photographyEditorial Note: This manuscript has been previously reviewed at another journal that is not operating a transparent peer review scheme. This document only contains reviewer comments and rebuttal letters for versions considered at *Nature Communications*.

REVIEWERS' COMMENTS

Reviewer #1 (Remarks to the Author):

The response letter of the authors did completely answer my questions and criticisms. I think that manuscript is perfectly suitable for publication in Nature Communications in the present form. The authors very convincingly (and elegantly) answered the main question concerning the manuscript's novelty, with respect to previous methods (point #2).

I agree with the modifications that have been made in the Manuscript (that were short and localized, as we could expected). I also appreciated the new Supplementary Note 2, This may indeed answer automatically the questions that the reader may have about the "fundamental limits" of this new method.

Note that the potential impact of the article is – from my best knowledge of single-shot THz techniques – potentially really high, as explained in my previous review (and also reminded in the first part of the author's Response).

For these reasons (the expected impact, originality/novelty of the method, and the technical quality of the research) I think that the article deserves publication in a high-level journal such as Nature Communications.

Response to reviewers' comments

We marked the Reviewers' comments in blue and our response in red. Modifications of the manuscript and supplementary material are highlighted in red.

Reviewer #1 (Remarks to the Author):

The response letter of the authors did completely answer my questions and criticisms. I think that the manuscript is perfectly suitable for publication in Nature Communications in the present form. The authors very convincingly (and elegantly) answered the main question concerning the manuscript's novelty, with respect to previous methods (point #2).

I agree with the modifications that have been made in the Manuscript (that were short and localized, as we could expect). I also appreciated the new Supplementary Note 2, This may indeed answer automatically the questions that the reader may have about the "fundamental limits" of this new method.

Note that the potential impact of the article is – from my best knowledge of single-shot THz techniques – potentially really high, as explained in my previous review (and also reminded in the first part of the author's Response).

For these reasons (the expected impact, originality/novelty of the method, and the technical quality of the research) I think that the article deserves publication in a high-level journal such as Nature Communications.

We thank the Reviewer for the careful assessment of our revised manuscript. We are glad that our responses and modifications applied to the manuscript have fully addressed the previous concerns expressed by the Reviewer. We believe that the Reviewer's comments and suggestions helped us significantly improve our manuscript. We appreciate her/his recommendation for publication in Nature Communications.

Prof. Roberto Morandotti

Nonlinear Photonics Group, INRS-EMT

1650 Blvd. Lionel-Boulet,

Varenes, Québec J3X 1S2, Canada